# Assessing Clinical Potential of Old Antibiotics against Severe Infections by Multi-Drug-Resistant Gram-Negative Bacteria Using In Silico Modelling

**DOI:** 10.3390/ph15121501

**Published:** 2022-11-30

**Authors:** Paschalis Paranos, Sophia Vourli, Spyros Pournaras, Joseph Meletiadis

**Affiliations:** 1Clinical Microbiology Laboratory, Attikon University Hospital, Medical School, National and Kapodistrian University of Athens, 12462 Athens, Greece; 2Department of Medical Microbiology and Infectious Diseases, Erasmus MC, 3015 CN Rotterdam, The Netherlands

**Keywords:** multi-drug-resistant bacteria, life-threatening infections, old antibiotics, Monte Carlo, probability of target attainment

## Abstract

In the light of increasing antimicrobial resistance among gram-negative bacteria and the lack of new more potent antimicrobial agents, new strategies have been explored. Old antibiotics, such as colistin, temocillin, fosfomycin, mecillinam, nitrofurantoin, minocycline, and chloramphenicol, have attracted the attention since they often exhibit in vitro activity against multi-drug-resistant (MDR) gram-negative bacteria, such as *Escherichia coli*, *Klebsiella pneumoniae*, *Pseudomonas aeruginosa*, and *Acinetobacter baumannii*. The current review provides a summary of the in vitro activity, pharmacokinetics and PK/PD characteristics of old antibiotics. In silico modelling was then performed using Monte Carlo simulation in order to combine all preclinical data with human pharmacokinetics and determine the probability of target (1-log kill in thigh/lung infection animal models) attainment (PTA) of different dosing regimens. The potential of clinical efficacy of a drug against severe infections by MDR gram-negative bacteria was considered when PTA was >95% at the epidemiological cutoff values of corresponding species. In vitro potent activity against MDR gram-negative pathogens has been shown for colistin, polymyxin B, temocillin (against *E. coli* and *K. pneumoniae*), fosfomycin (against *E. coli*), mecillinam (against *E. coli*), minocycline (against *E. coli*, *K. pneumoniae*, *A. baumannii*), and chloramphenicol (against *E. coli*) with ECOFF or MIC_90_ ≤ 16 mg/L. When preclinical PK/PD targets were combined with human pharmacokinetics, Monte Carlo analysis showed that among the old antibiotics analyzed, there is clinical potential for polymyxin B against *E. coli*, *K. pneumoniae*, and *A. baumannii*; for temocillin against *K. pneumoniae* and *E. coli*; for fosfomycin against *E. coli* and *K. pneumoniae*; and for mecillinam against *E. coli*. Clinical studies are needed to verify the potential of those antibiotics to effectively treat infections by multi-drug resistant gram-negative bacteria.

## 1. Introduction 

The ever-increasing antimicrobial resistance worldwide poses an urgent threat to our antimicrobial arsenal. In particular, gram-negative bacteria, such as *Escherichia coli*, *Pseudomonas aeruginosa*, *Klebsiella pneumoniae*, and *Acinetobacter baumannii*, rapidly develop resistance to many currently licensed antimicrobial drugs, including recently introduced drugs. According to the ECDC 2022 report, only 46% of *E. coli*, 62% of *K. pneumoniae*, 70% of *P. aeruginosa*, and 34% of *A. baumannii* were fully susceptible to all antimicrobial drugs [1]. Resistance rates to at least two antimicrobial groups were 10.2% for *E. coli*, 15.7% for *K. pneumoniae*, 7.6% for *P. aeruginosa*, and 57.3% for *A. baumannii*. The rates of multi-drug resistance (MDR) phenotypes are increasing every year, limiting therapeutic options. Of particular concern are extended-spectrum β-lactamase-producing *Enterobacterales*, carbapenem-resistant *A. baumannii* and *Enterobacterales*, and MDR *P. aeruginosa*, with the pipeline of new antimicrobial agents being very limited due to the time-consuming process and the exorbitant costs for the development of new and hopefully potent drugs [2]. As some old antibiotics introduced in 1950s show in vitro activity against those MDR bacteria, there has been an increased interest in those compounds as an alternative approach to treating MDR infections [3,4,5].

Among the old antimicrobial agents, some were neglected because new drugs have been introduced with improved activity and safety profiles and convenient administration routes. Among them, the most interesting compounds from a clinical point of view were considered to be colistin, temocillin, mecillinam, nitrofurantoin, fosfomycin, minocycline, and chloramphenicol [6]. Those drugs never underwent the processes that new drugs now undergo for drug efficacy assessment and regulatory approval. These agents are both cheap with broad spectrum activity, and some of them have been used successfully to treat non-severe infections, such as urinary tract infections, by MDR. One of the major advantages of old antibiotics is the fact that they are not currently widely used, and therefore, resistance levels are expected to be low [7].

However, data regarding the probability of therapeutic success and the appropriate dose for severe infections, such as bloodstream infections and pneumonia are sparse. It is imperative to assess these drugs based on more up-to-date pharmacokinetic/pharmacodynamic (PK/PD) studies and explore any potential for use against MDR infections. Old drugs were only recently assessed based on current guidelines on the use of pharmacokinetics (PKs) and pharmacodynamics (PDs) for the development of antimicrobial medicinal products. In silico modelling is an important tool for assessing whether licensed or alternative dosing regimens of those drugs can attain preclinical PK/PD targets [8]. We therefore reviewed in vitro susceptibility, PK, and PK/PD data of old drugs and performed in silico modelling in order to estimate the probability of target attainment (PTA) ousing previously published PK/PD targets.

## 2. Methods

In vitro susceptibility, PK and PK/PD data of colistin, temocillin, mecillinam, nitrofurantoin, fosfomycin, minocycline, and chloramphenicol against *E. coli*, *K. pneumoniae*, for *P. aeruginosa*, and *A. baumannii*—particularly MDR pathogens—were reviewed. Monte Carlo simulation was then performed in order to bridge in vitro susceptibility data, preclinical PK/PD targets, and human PKs and determine the PTA for each drug and gram-negative MDR species. Monte Carlo simulation analysis is a well-established approach to simulating exposures in a large number of patients and associated variation based on mean and standard deviation (SD) PK parameters derived from a small cohort of patients in clinical PK studies. For this reason, we used the KinFun 1.02 software (Maastricht, Netherlands) with the following input parameters: N, number of compartments; fu, unbound fraction; V1, volume central compartment; k10, (CL/V1) elimination rate constant; and k12 (Q/V1) and k21 (Q/V2), distribution rate constants, where Q is the intercompartmental clearance, V2 volume peripheral compartment, CL clearance. Rate constants were extracted from the literature; if there were no published reports, they were calculated based on reported CL and volume of distribution (Vd) values. SDs were used to add variability in CL and Vd. A log normal distribution was assumed for both CL and Vd in order to calculate individual rate constants for 5000 patients. Apart from that, route of administration, maintenance dose, number of doses, dosing time intervals, and infusion duration were taken into account in order to simulate PK profiles. Simulated CL, VD, and their corresponding SDs were compared to published values, allowing <3% deviation for all parameters. For each simulated patient, the PK/PD indices *f*AUC/MIC (area under the unbound plasma concentration time curve over the minimum inhibitory concentrations MIC), *f*Cmax/MIC (peak of unbound plasma concentration over MIC), and %*f*T > MIC (% of dosing interval that unbound plasma concentrations remains above the MIC) were calculated for different MICs, taking into account the unbound fraction of drugs, as this fraction is considered pharmacodynamically important. Furthermore, the number of simulated patients attaining preclinical PK/PD targets corresponding to 1-log kill were plotted against MICs together with the MIC distribution from the European Committee on Antimicrobial Susceptibility Testing (EUCAST) website (www.eucast.org accessed on 23 October 2022) and, when not available, from previously published papers. The 1-log kill effect was chosen as it is the most relevant endpoint for severe infections, such as bloodstream infections and pneumonia [9]. Therefore, only preclinical data from thigh and lung infection models were utilized for determining PK/PD targets and when not available, data from in vitro dynamic models were used. The PK/PD target of 3-log kill was used from in vitro models as this seems to correlate with 1-log kill in animals [10]. 

The process consisted of 1. simulation of a virtual patient population infected with pathogens with increasing MICs, 2. calculation of PK/PD indices for each patient and each pathogen, 3. estimation of % of patients attaining the PK/PD target of 1-log kill effect for each MIC, 4. comparison of the MICs with a PTA > 95% with the epidemiological cutoff value (ECOFF), or MIC_90_ when ECOFF was not available for each species. In addition, the cumulative fraction of response (CFR) [11] was estimated for each antibiotic dosing regimen against MIC distributions of *Enterobacterales*, *P. aeruginosa*, and *A. baumannii*, as presented on the EUCAST website (www.eucast.org accessed on 14 October 2022) or in previously published papers. CFR was defined as the cumulative PTA of an expected population for a specific dosing regimen and a specific population of microorganisms and was calculated based on the equation CFR = *Σ*PTA_i_xF*i* [11]: where the subscript i indicates the MIC value, ranked from highest to lowest MIC of the tested population of microorganisms; PTA_i_ is the probability of target attainment of each MIC; and F*i* is the fraction of microorganism population at each MIC category. 

A drug was considered promising against a specific pathogen when the PTAs were >95% for the wild-type (WT) population, i.e., at the ECOFF (or the lowest concentration of antibiotic at which 90% of the isolates were inhibited (MIC_90_) if ECOFF has not been determined). 

## 3. Colistin

Colistin (polymyxin E) is a member of the polymyxin group of antimicrobial compounds, which consist of basic polypeptide antibiotics with a side chain terminated by fatty acids. Colistin and polymyxin B were introduced to the pipeline of antibiotics for clinical use with similar antibacterial spectra. Colistin acts by displacing calcium and magnesium from the negatively charged lipopolysaccharide in the cell membrane of bacteria, leading to increased permeability of the cell envelope, loss of integrity of the membrane, leakage of cell contents, and finally, cell death. Acquired resistance to colistin is mainly due to lipopolysaccharide modifications [12]. 

Colistin is highly active against *Enterobacterales*, except *Proteae* and *Serratia* spp. It is also active against *Pseudomonas aeruginosa* and *Acinetobacter spp*. Based on EUCAST MIC distribution, colistin is highly potent against various gram-negative bacteria, with ECOFFs of 2, 2, 4, and 2 mg/L for *E. coli*, *K. pneumoniae*, *P. aeruginosa*, and *A. baumannii*, respectively (www.eucast.org accessed on 23 October 2022) (Table 1). 

Colistin is administered to patients intravenously (IV) as the inactive prodrug colistin methanesulphonate sodium (CMS), which is converted to its active form in vivo. It is used for the treatment of infections caused by isolates that are resistant to less toxic antimicrobials. Moreover, patients with chronic bronchopulmonary colonization by *P. aeruginosa* are treated with colistin in inhaled form [12]. The most often used clinical IV dosing regimens are 9MIU q24h, 4.5MIU q12h, and 3MIU q8h, with tAUC_0–24_ 50.18–72.93 mg∙h/L and tCmax 2.98–5.83 mg/L [39] (Table 2). The protein binding of colistin in plasma of 66 critically ill patients (and healthy humans) was found to be 40% (unbound fraction 0.5) [40]. Large fluctuations have been found in t_1/2_ ranging between 2–14.4 h [41,42]. 

The PK/PD index linked to antibacterial effect is the ratio *f*AUC/MIC [39]. *f*AUC/MIC values of 20.37 ± 4.13 and 14.83 ± 2.35 were previously found to correspond to 1-log kill of *P. aeruginosa* in thigh and lung infection animal models, respectively [58] (Table 3). Lower PK/PD indices were found in another study in thigh infection animal models (6.6–10.9 and 3.5–13.9 *f*AUC/MIC was associated with 1-log kill of *P. aeruginosa* and *A. baumannii*, respectively), probably because the unbound fraction for colistin was estimated to be lower (concentration-independent fu 0.084) compared to the previous studies (concentration-dependent fu 0.1–0.5). Thus, the PK/PD target for *A. baumannii* was underestimated, and for that reason, we used the strictest target found for the study above (*f*AUC/MIC 13.9) [40]. Regarding *K. pneumoniae*, in an In vitro PK/PD model, an *f*AUC/MIC 24 corresponded to a bactericidal effect of 3-log kill [39]. 

Monte Carlo simulation of 5000 patients was performed with a mean ± SD *f*AUC_0–24_ of 29.17 ± 15.44 for 9MU q24h (45 min IV infusion), 24.28 ± 4.8 for 4.5MU q12h (45min IV infusion), and 20.07 ± 5.37 for 3MU q8h (45 min IV infusion) [43] and using the mean PK/PD indices 20.37 for *P. aeruginosa*, 13.9 for *A. baumannii*, and 24 for *K. pneumoniae*. The PTAs were low at ECOFF of 4, 2 and 2 mg/L for *P. aeruginosa*, *A. baumannii* and *K. pneumoniae* for all dosing regimens tested, respectively (Figure 1). Considering the same PK/PD target of *f*AUC/MIC 24 as in *K. pneumoniae*, PTAs were <10% at ECOFF 2 mg/L for *E. coli* isolates for all dosing regimens tested. Hence, colistin seems not be a promising agent as monotherapy at the abovementioned doses against the MDR gramm(-)bacteria. This was confirmed by an open-label randomized controlled trial study; clinical failure rates ranged between 62–83% in patients treated with colistin monotherapy against infections caused by *Enterobacterales*, *P. aeruginosa*, and *A. baumanii* [66]. 

## 4. Polymyxin B

Polymyxin B is a cationic polypeptide antibiotic with very similar chemical structure to colistin (Polymyxin E), and it is mainly used for the treatment of infections caused by gram-negative bacteria—in particular, MDR *K. pneumoniae*, *A. baumannii*, and *P. aeruginosa* [67]. The mechanism of action involves the ability to bind with and disorganize the outer membrane of gram-negative bacteria, disrupting the osmotic equilibrium [68]. Resistance mechanisms include intrinsic and adaptive resistance via mutations of LPS, whereas horizontally acquired resistance mechanisms have not been reported [68]. Based on previous studies, polymyxin’s B MIC_90_ was 0.25 mg/L for *E. coli*, 0.5 mg/L for *K. pneumoniae*, 2 mg/L for *P. aeruginosa*, and 0.25–0.5 mg/L for *A. baumannii* isolates (Table 1) [18,22,23]. Higher polymyxin B MIC_90_ values were observed for carbapenem-resistant (CR) *K. pneumoniae* (MIC_90_ 2–32 mg/L), although no information on previous treatment and resistance mechanisms was provided [20].

Polymyxins are very large lipopeptide molecules and are poorly absorbed via oral administration [69]. They are commonly given IV for the treatment of life-threatening systemic infections or by nebulization for the treatment of respiratory tract infections [70]. Pharmaceutical products contain active polymyxin B as sulphate salt [67]. Although polymyxin B is a potent antimicrobial compound, its clinical utility is widely limited by its potential for nephrotoxicity and neurotoxicity [28,71]. In general, dosage is based on total body weight, but PK data are not explored in all group of patients [72]. Due to significant knowledge gaps, the clinical dosing strategies of polymyxin B are not fully optimized, and for that reason, a thorough understanding of PKs is crucial to maximizing efficacy and minimizing toxicity [73]. 

The current understanding of polymyxin B PKs consists of four studies comprised of total 60 patients in total [74]. Clinically administered doses usually result with maximum serum concentration at steady state ranging from 2–14 μg/mL with a half-life of 9–11.5 h [75]. PKs of polymyxin B in different group of patients at clinical dosing regimens of 40–50 mg q12h, 119 ± 36.3 mg/day and 0.45–3.38 mg/kg/day resulted in mean ± SD AUC_0–24_ values of 74.6 ± 17.81, 52.3 ± 14.8, and 66.9 ± 21.6 mg·h/L, respectively (Table 2). The unbound fraction of polymyxin B in plasma of 24 critically ill patients was found to be 42% [46]. Half-life of the drug (t_1/2_) ranged between 10.1 and 11.9 h. 

According to the findings of previous PK/PD studies, the PK/PD index that best describes the bactericidal activity of polymyxin B is AUC_0–24_/MIC [44]. *f*AUC/MIC values of 39.7 ± 14.4 and 50.6 ± 3.8 were associated with 1-log kill of *K. pneumoniae* and *E. coli*, respectively, in a thigh infection animal model (Table 3) [60]. Monte Carlo simulation analysis of 5000 patients was performed with mean ± SD *f*AUC_0–24_ 31.33 ± 7.48, 21.96 ± 6.21, and 28.09 ± 9.07 for dosing regimens of 40–50 mg q12h, 119 ± 36.3 mg/day and 0.45–3.38 mg/kg/day, respectively. The PTA was >95% with the dosing regimen of 40–50 mg q12h in renal transplant patients for most *E. coli* (MIC_90_ 0.25 mg/L) and *K. pneumoniae* (MIC_90_ 0.5 mg/L) isolates (Figure 2). Considering the same PK/PD target for the other gram-negatives, the PTAs are expected to be high (>95%) for most *A. baumannii* isolates, as their MIC_90_ is ≤0.5 mg/L, but not for *P. aeruginosa* (MIC_90_ 2 mg/L) or CRE *K. pneumoniae* (MIC_90_ 2-32 mg/L) for all three dosing regimens tested. Indeed, IV polymyxin B therapy was inferior to other drugs in the treatment of *P. aeruginosa* bacteremia and *K. pneumoniae* [76,77], whereas the use of polymyxin B was effective against *A. baumannii* infections [78]. For that reason, polymyxin B seems to be quite promising against *E. coli*, *K. pneumoniae*, and *A. baumannii* but not against *P. aeruginosa* infections. Studies assessing clinical efficacy of polymyxin B against multi-drug-resistant gram-negative bacteria found lower mortality for *A. baumannii* than *P. aeruginosa* infections, although in most cases, polymyxin B was given in combination with other drugs [78,79].

## 5. Temocillin

Temocillin, a 6-a-methoxy derivative of ticarcillin, is a penicillin with restricted spectrum against *Enterobacterales*, while non-fermenters, anaerobes and gram-positive bacteria are not within its spectrum [80]. The addition of α-methoxy moiety on ticarcillin prevents hydrolysis against a wide variety of β-lactamases, including extended spectrum β-lactamases (ESBLs) [81], ampicilinases C (AmpCs) [3] and some carbapenemases [82], but not metallo-beta-lactamases (MBL) and oxacillininases (OXAs) [3,29]. The mechanism of action relies on the prevention by α-methoxy group of entry of a water molecule into the β-lactamase active site, which leads to hydrolysis by preventing activation of the serine and other chemical events [3]. On the other hand, the intrinsic resistance of *P. aeruginosa* isolates to temocillin is mainly due to active efflux by the constitutively expressed MexAB-OprM efflux transporters [83]. In addition, there are in vitro studies that demonstrate the efficacy of active drugs against ESBL-producing strains, with susceptibility rates of up to 80% and 90% using breakpoints of 8 and 16 mg/L, respectively [26,81,84,85]. In addition, an observational study supports the clinical use of temocillin as a potential alternative to carbapenems against infections caused by ESBL-/AmpC-producing *Enterobacterales* [86]. Based on EUCAST distribution, temocillin is highly potent against various gram-negative bacteria, with ECOFF being 16 and 8 mg/L for *E. coli* and *K. pneumoniae*, respectively (www.eucast.org). The MIC_90_ of temocillin was ≥256 mg/L against *P. aeruginosa* and *A. baumannii* (Table 1) [83,87].

Temocillin can only be administered parenterally (IV, intramuscularly, and subcutaneously), and it is used for empirical treatment of pyelonephritis and complicated UTI [57]. It is bactericidal, has a prolonged elimination half-life of approximately 5 h, and has a high percentage of protein binding (~80%). Clearance of temocillin is mainly renal, and urinary recovery ranges from 72–82% after 24 h. Moreover, it has high penetration into bile and peritoneal fluid but poor penetration into cerebrospinal fluid. PKs of temocillin were simulated in different dosing regimens and different groups of patients, as shown in Table 2. Briefly, clinical dosing regimens of 2 g q12h (4 g/24 h) and 2 g q8h (6 g/24 h) result in AUC_0–24_ values of 1856 ± 282 and 1764 mg·h/L, and C_max_ values of 147 ± 12 and 170 mg/L, respectively. The percentage of free drug was 23.7 ± 6.15% [47]. The half-life of the drug was 4.3 ± 0.3 h.

The PK/PD index correlating with temocillin efficacy seems to be *f*T > MIC. %fT > MIC values of 81.5 ± 14.4 and 79 ± 6.4 were associated with 1-log kill of *E. coli* and *K. pneumoniae* in thigh and 35 ± 18.3 and 47.3 ± 21.4 for lung infection animal models, respectively [61] (Table 3). Monte Carlo simulation of 5000 patients with mean ± SD Cl 2.44 ± 0.39 L/h and VD 14.3 ± 0.87 L for 2g q12h, and with mean ± SD Cl 3.69 ± 0.45 L/h, V1 14 ± 2.51 and V2 21.7 ± 4.52 for 2g q8h was performed using KinFun. For the Monte Carlo analysis, an unbound fraction of 24% has been used for both dosing regimens. For *E. coli*, the PTAs of the 1-log kill PK/PD target in the thigh infection model were low (<50%) at the ECOFF of 16 mg/L with the 2g q12h and 2g q8h dosing regimens, whereas the PTAs of 1-log kill PK/PD target in lung infection model were >90% at the ECOFF of 16 mg/L with a 2g q8h dosing regimen (Figure 3). For *K. pneumoniae*, the PTA of the PK/PD target in thigh infection model was <85% at the ECOFF of 8 mg/L for both dosing regimens, whereas the PTA of the PK/PD target in the lung infection model was 100% at the ECOFF of 8 mg/L with both dosing regimens. If the PK/PD targets were the same for *P. aeruginosa* and *A. baumannii*, PTAs are expected to be low for WT isolates with 2g q12h and 2g q8h dosing regimens due to high MIC_90_ (≥256 mg/L) for both species [29]. Providing that the same tissue penetration occurs in humans, the 2g q12h and 2g q8h dosing regimens are promising for pneumonia by *E. coli* and *K. pneumoniae*. 

In a large, multicenter, retrospective, open, noncomparative study, clinical/microbiological efficacies of temocillin against infections by *Enterobacterales* (55% *E. coli*, 14% *K. pneumoniae*) were 83%/82% for BSI and 75%/67% for hospital-acquired pneumonia (HAP), being slightly lower than the 90%/87% in urinary tract infection (UTI) [86]. In a single-center, retrospective study, clinical failure was higher in non-UTI than UTI (26.7% vs. 4.9%) and in patients with septic shock compared to patient with sepsis (25% vs. 6.2%) by *Enterobacterales* (mainly *E. coli*), with no difference observed between the 2g q12h and 2g q8h dosing regimens, although q8h regimen was more frequently given in seriously ill patients [88]. Interestingly, significantly fewer failures were observed for *K. pneumoniae* infections. Thus, temocillin may have a role in treating *E. coli* and *K. pneumoniae* infections.

## 6. Fosfomycin

Fosfomycin, first isolated from *Streptomyces* spp. in 1969, is an organic phosphonate agent. It is the smallest molecule among all antimicrobials (138 Da), and cross-resistance with other classes is rare. It is rapidly bactericidal, inhibiting cell wall synthesis via irreversible inhibition of the enol-pyruvyl transferase. Penetration into the bacterial cell occurs through two different active transport systems, the inducible and predominant hexose monophosphate route which operates in the presence of glucose-6-phosphate inducer (GlpT) and the constitutive L-α-glycerophosphate system (UhpT) [89]. Concerning resistance mechanisms to fosfomycin, there are two intrinsic and one acquired mechanisms. For intrinsic mechanisms, inactivation of fosfomycin occurs via cleavage of the molecule by bacterial Fos enzymes. The majority of *K. pneumoniae*, *K. aerogenes*, *P. aeruginosa*, and *Enterobacter* spp. have FosA enzymes. Moreover, fosfomycin inhibits MurA, which initiates peptidoglycan biosynthesis of the bacterial cell wall. Resistance to fosfomycin in several bacteria is common mainly through *MurA* mutations, to which fosfomycin must bind to exhibit its antibacterial effect. Acquired resistance to fosfomycin occurs through modifications of membrane transporters GlpT and UhpT preventing active drug entering the bacterial cell, resulting in reduced uptake of fosfomycin by the pathogen [89]. Although resistance rates in clinical isolates are still relatively low, the emergence of resistance occurs rapidly in vitro. Resistant mutants arise in vitro at a frequency of 10^−4^ to 10^−5^. In ESBL-producing *E. coli*, in vivo resistance is increasingly recognized [90].

Fosfomycin has a broad spectrum of activity including both gram-negative and gram-positive bacteria and recently gained considerable attention due to its effectiveness against multi-drug-resistant pathogens [91], including ESBL and carbapenemase-producing isolates. Based on EUCAST distribution, fosfomycin is highly potent against *Enterobacterales* but seems less effective against *P. aeruginosa* and *A. baumannii* isolates. The ECOFF for *E. coli*, *K. pneumoniae*, and *P. aeruginosa* are 4, 128, and 256 mg/L, respectively (www.eucast.org, accessed on 23 October 2022). High MIC_90_ (>256 mg/L) has been reported for carbapenemase-producing *Enterobacterales* (Table 1). Fosfomycin resistance among carbapenemase-producing *Enterobacterales* is an emerging problem and is due to the plasmid-mediated fosfomycin resistance gene fosA3 and mutation in the transporter glpT [92].

Fosfomycin is hydrophilic with negligible protein binding and is eliminated exclusively by glomerular filtration, so its clearance depends mainly on the patient’s renal function. Regarding the volume of distribution, it is approximately 0.3 L/kg, but in critically ill patients suffering from bacterial infections, it is increased [93]. Moreover, fosfomycin is well tolerated and exhibits extensive penetration into many tissues with minor side effects reported [94]. PKs of fosfomycin were simulated in non-critically-ill patients based on the most recent study, as is shown in Table 2. The clinical dosing regimen of 4 g q6h resulted in a mean ± SD fAUC_0–24_ values of 5215 ± 1972.2 mg∙h/L the first 24 h [50] although lower exposures have been described in other studies [90]. Fosfomycin does not bind to plasma proteins [90]. The half-life of the drug ranged between 2.41–12.1 h depending on the route of administration [91].

Fosfomycin is hydrophilic with negligible protein binding and is eliminated exclusively by glomerular filtration, so its clearance depends mainly on the patient’s renal function. Regarding the volume of distribution, it is approximately 0.3 L/kg, but in critically ill patients suffering from bacterial infections, it is increased [93]. Moreover, fosfomycin is well tolerated and exhibits extensive penetration into many tissues with minor side effects reported [94]. PKs of fosfomycin were simulated in non-critically-ill patients based on the most recent study, as is shown in Table 2. The clinical dosing regimen of 4 g q6h resulted in a mean ± SD fAUC_0–24_ values of 5215 ± 1972.2 mg∙h/L the first 24 h [50] although lower exposures have been described in other studies [90]. Fosfomycin does not bind to plasma proteins [90]. The half-life of the drug ranged between 2.41–12.1 h depending on the route of administration [91].

The optimal PK/PD index characterizing fosfomycin activity is AUC/MIC, as found in dose fractionation studies in animals (R^2^ = 0.70 for AUC/MIC, R^2^ = 0.51 for Cmax/MIC, R^2^ = 0.44 for T > MIC) [62]. Monte Carlo simulation analysis of 5000 patients was performed for 4 g q8h (12 g/d), 6 g q8h (18 g/d), and 8 g q8h (24 g/d). The *f*AUC_0–24_ was calculated based on the *f*AUC_0–24_ of 5215 ± 1972.2 mg∙h/L for the 4g q6h (16 g/d) as described by Merino-Bohorquez et al. [50], assuming linear PKs and the same variation across different total daily doses, with mean ± SD *f*AUC_0–24_ 3911 ± 1479 mg∙h/L for 4 g q8h (12 g/d) dosing regimen, 5867 ± 2219 mg∙h/L for 6 g q8h (18 g/d), and 7823 ± 2958 mg∙h/L for 8 g q8h (24 g/d). Mean ± SD AUC_0–24_/MIC values of 98.9 ± 78.4, 21.5, and 28.2 ± 17.82 corresponded to 1-log kill of *E. coli*, *K. pneumoniae*, and *P. aeruginosa*, respectively were previously found in thigh infection animal models [62] (Table 3). The PTAs were 100% at the ECOFF of 4 mg/L for *E. coli* for all simulated dosing regimens tested (Figure 4). This also can be confirmed by the high rate of CFR (>95%) in all dosing regimens. The PTAs were high (>95%) at the ECOFF of *K. pneumoniae* isolates only for dosing regimens with 18g/day, while for *Klebsiella pnemoniae* carbapenemase (KPC), New Delhi metallo-β-lactamase (NDM), MBL, OXA-48-producing isolates with MIC_90_ 256 mg/L, and for WT *P. aeruginosa* (MIC ≤ 256 mg/L) isolates, none of the dosing regimens could be effective (PTA ≤ 74%). Fosfomycin has been rarely used as monotherapy to treat severe infections by MDR pathogens. In a multi-center, randomized, double-blind comparative study (ZEUS study), fosfomycin was given as monotherapy mainly against UTI, and clinical cure was observed in 25/27 infections caused by *K. pneumoniae*, 8/8 caused by *P. aeruginosa*, 120/133 caused by *E. coli*, and 2/2 caused by *A. baumannii* [95]. Thus, there is a clinical potential for fosfomycin against *E. coli* and *K. pneumoniae*, although the high MICs reported for carbapenemase-producing *Enterobacterales* may limit fosfomycin coverage.

## 7. Mecillinam

Mecillinam, or 6β-amidinopenicillanic acid, is an amidinopenicillin developed in 1972 and has been used extensively in Scandinavian countries for the treatment of acute lower UTI caused by *Enterobacterales*, and especially by *E. coli*, since the 1980s [96]. The antimicrobial is detected in high concentrations in urine, and its impact on the intestinal microbiota was found to be low [97,98]. The antimicrobial agent is administered orally as pivmecillinam, which is hydrolyzed to the active drug in vivo. The prodrug pivmecillinam is a unique β-lactam with high specificity against penicillin-binding protein 2 (PBP-2) in gram-negative cell walls, and extensive activity against *Enterobacterales*, that also resists hydrolysis by β-lactamases [5]. Biochemical and genetic studies revealed that mecillinam interacts with PBP-2, resulting in the production of round, osmotically stable bacterial cells [99]. Resistance development is associated with mutations in a large number of genes that affect many different cellular functions, including cell elongation and division, composition of lipopolysaccharide in combination with cya/crp, and cysteine biosynthesis [100]. Evaluating the in vitro efficacy of the drug, remarkable activity is retained against ESBL and AmpC β-lactamases. Moreover, recent data suggest that mecillinam is frequently active in vitro against NDM and imipenemase (IMP) metallo-β-lactamases and OXA-48 producers but not against KPC and Verona integron-encoded metallo-β-*lactamase* (VIM) [3,100]. Based on EUCAST MIC distribution, mecillinam seems to be potent against *Enterobacterales* isolates. The MIC_90_ values were 2 mg/L and 128 mg/L for *E. coli* and *K. pneumoniae*, respectively, with a tentative ECOFF of 0.5 mg/L for *E. coli* (Table 1). Mecillinam is inactive against *P. aeruginosa* and *A. baumannii* (MIC > 128 mg/L) [101,102].

Pivmecillinam has high bioavailability (~70%), with 45% of the dose being secreted in urine as mecillinam within 6 h of administration [103]. Side effects are rare, with the most common being mild gastrointestinal symptoms [104]. The use of pivmecillinam as treatment for uncomplicated UTI is recommended by the European Society for Clinical Microbiology and Infectious Diseases, the European Association of Urology, and the Infectious Diseases Society of America [105]. Serum PK studies of 10 mg/kg mecillinam in healthy patients resulted in a mean C_max_ of 61 mg/L (Table 2). There are no PK data from critically ill patients. The unbound fraction of the drug was calculated to be 90–95% [106]. The half-life of the drug was calculated to be 0.5 h. 

Monte Carlo simulation of 5000 patients was performed with mean ± SD Cl 14.7 ± 1.4 L/h and VD 16.1 ± 2.8 L for 1 g q8h and 1 g q6h. As there are no PK/PD studies for mecillinam, the 50%*T* > MIC corresponding to 1-log kill against *E. coli* was used as found for most penicillins [57] (Table 3). The PTAs for the latter target were high (99%) for the tentative ECOFF of 0.5 mg/L for *E. coli* with the dosing regimen of 1 g q6h (Figure 5). PTAs are low for both dosing regimens for *K. pneumoniae* isolates as MIC_90_ (128 mg/L) is six-fold higher than *E. coli* according to EUCAST MIC distribution (Table 1). In the few patients where mecillinam was used against bacteremia, clinical and bacteriological success rates were 67% (10/15) and 87% (13/15), respectively [103,107]. Thus, mecillinam is a promising agent for treating gram-negative infections caused by *E. coli* at the dose of 1 g q6h.

## 8. Nitrofurantoin

Nitrofurantoin was introduced in clinical practice in 1953 [108] and is the most widely used antimicrobial within the nitrofuran class. Moreover, it is the only member of the nitrofuran family that is in use in human medicine and is available only as an oral formulation [109]. It is an old antibiotic that has been used for the treatment of uncomplicated UTI for decades, and its consumption increased as a first-line agent for the treatment of cystitis after the guidelines were updated in 2011 [105]. At low concentrations, nitrofurantoin inhibits the inducible synthesis of β-galactosidase and galactokinase without affecting total protein synthesis, while at higher concentrations, it inhibits enzymes of the citric acid cycle as well as DNA, RNA, and total protein synthesis in bacteria via a mechanism involving the reaction of electrophiles following bacterial reduction of nitrofurantoin with nucleophilic sites on bacterial macromolecules [110]. Concerning resistance mechanisms for nitrofurantoin, several mechanisms have been proposed, including mutations in *nfsA* and *nfsB* genes as well as the presence of the *oqxAB* gene [111]. Nitrofurantoin is mainly bacteriostatic but can also exhibit bactericidal effects when present at high concentrations (≥2 × MIC) [63,112]. Among the advantages of nitrofurantoin are the low prevalence of resistance amongst *Enterobacterales* and the low repercussions in commensal flora in comparison to the impact of quinolones or β-lactams [113,114]. Despite its extensive use, resistance rates are still low [115]. Its spectrum of activity includes ESBL-producing *Enterobacterales*—with the exception of *Klebsiella* and *Proteae* strains (e.g., *Proteus*, *Morganella*, *and Providencia* spp.), which show intrinsic resistance—*Staphylococcus saprophyticus*, and vancomycin-resistant enterococci [112,116,117]. Based on EUCAST MIC distribution, nitrofurantoin seems to be more potent against *E. coli* isolates, with an ECOFF of 64 mg/L (Table 1). Low susceptibility rates (MIC ≤ 32 mg/L) were found for *K. pneumoniae* (37.9%), *P. aeruginosa* (8%), and *A. baumannii* (8.3%) [118]. 

Following oral administration, nitrofurantoin is excreted rapidly via the kidney, resulting in high urine and low serum concentrations. The formulations of nitrofurantoin have been changed over the years, and currently, the clinical regimens of 100 mg q8h and 50 mg q6h are the most common. Furthermore, PK properties differ significantly between the products, and this is mainly because nitrofurantoin is not a uniform product because of different crystal sizes of nitrofurantoin [109]. 

PKs of nitrofurantoin were simulated in different dosing regimens in healthy female patients, as shown in Table 2**,** due to absence of data from critically ill patients. Clinical dosing regimens of 50 mg q6h and 100 mg q8h had resulted in AUC_0–24_ values of 4.43 ± 0.96 and 6.49 ± 2.9 mg·h/L and C_max_ values of 0.326 ± 0.081 and 0.69 ± 0.35 mg/L, respectively. The unbound fraction of the drug was calculated to be between 25–50% [119]. The half-life of the drug ranged between 1.7–2.3 h.

The PK/PD index that best correlates with the drug’s antibacterial effect is still under investigation [120]. The PK/PD target of 82 %*f*T > MIC was associated with 3-log kill of *E. coli* isolates in an in vitro kinetic model [63], while no PK/PD study has been performed in animals (Table 3). Monte Carlo simulation of 5000 patients was performed with mean ± SD Cl 46.2 ± 18.6 L/h and VD 103.8 ± 65.9 L for a dosing regimen of 100 mg q8h and Cl 36.4 ± 11.4 L/h and VD 100 ± 49.6 L for 50 mg q8h [53]. The PTA for nitrofurantoin was low for *E. coli* with both dosing regimens (Figure 6). Oral nitrofurantoin has been used for UTI and pyelonephritis but not for the treatment of severe infections. Thus, oral nitrofurantoin is not promising for treating severe MDR gram-negative infections. 

## 9. Minocycline

Minocycline is a semisynthetic second-generation tetracycline that was first introduced in clinical practice in the 1970s. The IV formulation of the drug was withdrawn from the US market in 2005 due to decreased use but was reintroduced in May 2009 as an important option for the treatment of MDR organisms [71]. Minocycline is bacteriostatic and inhibits protein synthesis by binding to the 30 s ribosomal subunit [121]. A variety of mechanisms, including modification or protection of the antibiotic target site and efflux pumps, are involved in bacterial resistance to minocycline, such as Tet(B) and RND- type efflux pumps [122]. Minocycline has a broad spectrum of action against aerobic and anaerobic gram-negative and gram-positive bacteria, including some strains of streptococci, staphylococci, and *Haemophilus influenzae* resistant to tetracycline. Minocycline is also currently approved by the FDA in the United States for the treatment of infections caused by susceptible *A. baumannii* isolates [64]. Based on EUCAST MIC distribution, minocycline is active against *Enterobacterales* with ECOFF values of 4 and 8 mg/L for *E. coli* and *K. pneumoniae*, respectively (Table 1). Moreover, recent surveillance studies have shown that minocycline is potent against MDR and carbapenem-resistant *A. baumannii* with MIC_90_ 8 mg/L [71,123,124,125]. Regarding *P. aeruginosa*, a study demonstrated that 25% and 100% of isolates tested were inhibited by minocycline at a drug concentration of 25 and 50 mg/L, respectively [126].

Minocycline achieves excellent oral absorption and tissue penetration and a long elimination half-life, ranging from 15 to 23 h depending on administration of either 100 mg q12h or 200 mg q24h, respectively [127,128,129]. Most published PK data for IV minocycline concern healthy volunteers and are from studies conducted in 1970s [130]. Moreover, PK of minocycline has not been fully characterized in patients with creatinine clearance (CL_CR_) of <80 mL/min [129], and the FDA-approved product indicates that current data among patients with renal impairment are insufficient to determine if dose adjustments are warranted [130]. PK analysis of minocycline 200 mg resulted in an AUC_0–24_ of 24.3 ± 7.88 mg·h/L and a C_max_ of 2.58 ± 1.33 mg/L (Table 2). The unbound fraction of minocycline in plasma was found to be 30 ± 12% [54]. The half-life of the drug was calculated to be 1.36 ± 0.45 h. 

Monte Carlo simulation analysis of 5000 patients was performed with mean ± SD *f*AUC_0–24_ 7.29 ± 2.36 mg·h/L for the dosing regimen tested. The PK/PD index that best correlates with the drug’s antibacterial effect seems to be *f*AUC/MIC, according to several studies [54,64,131,132], with an *f*AUC_0–24_/MIC 21.08 associated with 1-log kill of *A. baumannii* in pneumonia infection animal models [64] (Table 3). The PTAs for 200 mg q24h of minocycline were <4% at MIC_90_ of *A. baumannii* (Figure 7). Since there were no PK/PD targets for *E. coli* and *K. pneumoniae*, we used the same target as *A. baumannii.* The PTAs are expected to be low at ECOFFs of 4 mg/L for *E. coli*, 8 mg/L for *K. pneumoniae*, and MIC_90_ 25–50 mg/L for *P. aeruginosa*. Successful treatment of MDR *A. baumannii* pneumonia with minocycline has been previously reported, although in most cases, minocycline was used in combination with other drugs and copathogens were involved [124]. Even with a higher exposure of *f*AUC_0–24_ of 25 mg·h/L previously reported [124], the PTA is high for isolates with MIC just up to 0.5 mg/L. Because human lung concentrations of minocycline are 4x plasma concentrations [128], if lung penetration in the rat model is lower, the PK/PD target must be adjusted. However, PTA will be still low even with higher exposure in lungs without covering all WT *A. baumannii* isolates (PTA 60% for MIC of 4 mg/L). Thus, minocycline is not a promising agent against MDR gram-negative infections.

## 10. Chloramphenicol

Chloramphenicol is the first broad-spectrum antibiotic to be manufactured synthetically on a large scale [133]. In most countries, it is available as topical agent and in some countries for parenteral administration. Nevertheless, due to rare but serious toxicity, it is now less used parenterally. Chloramphenicol binds to the 50S ribosomal subunit, inhibiting protein synthesis. Acquired chloramphenicol resistance comes from the production of the enzyme chloramphenicol acetyltransferase, but resistance can also be due to ribosomal modifications or altered permeability. It is active against various organisms, including gram-positive and gram-negative bacteria and anaerobes but less so against *Bacteroides* spp. Moreover, it is potent against *Mycoplasma* spp., *Rickettsia* spp., *Chlamydia* spp., and *Leptospira* spp. It is bacteriostatic but can be bactericidal at 2–4 × MIC against some gram-positive cocci, *Neisseria* spp., and *Haemophilus influenzae* [121]. Based on EUCAST MIC distribution, chloramphenicol’s ECOFF is 16 mg /L for *E. coli*, while the MIC_90_ was 8 mg/L for *K. pneumoniae* and ≥32 mg/L for MDR *A. baumannii* isolates (Table 1). Regarding other species, there is a study indicating that among MBL-positive isolates of *P. aeruginosa*, 68% were resistant to chloramphenicol according to CLSI guidelines [134]. Similar resistance rates (86.2%) were also found in a collection of MDR *K. pneumoniae* isolates carrying *intI*1 gene [135].

PKs of chloramphenicol were simulated in different dosing regimens and different groups of patients, as shown in Table 2. Briefly, clinical dosing regimens of mean 65.2 mg/kg/day and 1 g q6h resulted in AUC_0–24_ values of 468 ± 498 mg·h/L [55] and 518 mg·h/L [55], respectively, while single IV 30 mg/kg resulted in an AUC_0–∞_ of 72 ± 32 mg·h/L and a C_max_ of 16.2 ± 9.1 mg/L [124]. The protein binding of chloramphenicol in plasma was found to be ~40% [136]. The half-life of the drug was calculated to be 1.2 ± 1.15 h. 

PDs of chloramphenicol have never been studied in animal or human models. There is only one study that demonstrates the PDs of the related agent florfenicol, a derivative of chloramphenicol, indicating that AUC_0–24_/MIC is the PK/PD driver with an AUC_24_/MIC of 97.1 being associated with 1-log kill of *E. coli* in an ex vivo pig ileum model [65] (Table 3). Thus, Monte Carlo simulation analysis was performed with mean ± SD *f*AUC 187.2 ± 199.2 mg·h/L for a dosing regimen of 62.5 mg/kg/day, 28.8 ± 12.8 mg·h/L for 30 mg/kg/day and 207.2 ± 103.6 mg·h/L for 1 g q6h. The PTA was low at an ECOFF of 16 mg/L for *E. coli* for all dosing regimens (Figure 8). There are no clinical studies for the treatment of MDR gram-negative bacterial infections. Thus, chloramphenicol is not a promising agent.

## 11. Conclusions

In an era of increasing antibiotic resistance and scarcity of effective antimicrobials, there was a great interest on old antibiotics to combat MDR infections, as most of them demonstrated in vitro activity against these pathogens. Although they have been empirically used to treat infections—mainly mild infections but also more serious life-threatening infections in combination with other drugs—knowledge gaps on their PKs, PDs, and PK-PD characteristics and lack of clinical trials could not help to infer firm conclusion on their clinical efficacy [137]. Therefore, in recent years, the research community has made an enormous effort to fill in those gaps and explore the clinical potential of those antibiotics. In silico trials could help the efforts towards this direction, as modern tools of PK-PD analysis are utilized in order to determine whether old antibiotics could attain preclinical PK/PD targets in an approach used to assess new drugs during development and regulatory approval. The current review summarizes all in vitro MIC, PK, PK/PD, and PTA data in order to assess the clinical potential of old antibiotics to treat MDR infections and to provide hypotheses that need to be tested clinically. It has to be emphasized that in silico simulations conducted in the present analysis may not be extrapolated to other patient populations (e.g., critically ill, renal impairment, etc) as PKs may differ [138]. Furthermore, other factors, such as emergence of resistance, host factors, and infection sites where drugs may concentrate were not explored.

Among the old antibiotics tested, the most promising drugs were polymyxin B against *E. coli*, *K. pneumoniae*, and *A. baumannii*; temocillin against *K. pneumoniae* and *E. coli*; fosfomycin against *E. coli* and *K. pneumoniae*; and mecillinam against *E. coli*. As some MDR pathogens may have also high MICs to old antibiotics (polymyxin B and fosfomycin against CRE), the efficacy of the latter may be compromised against those pathogens. This emphasizes the importance of the careful use of old antibiotics with a clinical potential to treat MDR infections because irrational use can render them ineffective very quickly. The remaining antibiotics—colistin, minocycline, nitrofurantoin, and chloramphenicol—did not attain preclinical PK/PD targets in the current licensed forms and dosages (Table 4). 

However, even for drug–bug pairs with low probability of PK/PD target attainment, old antibiotics may have a role in treating MDR infections in combination with other drugs, as multiple in vitro studies demonstrated the synergism of old antibiotics when combined with other antimicrobials. For example, synergistic or additive activity were observed in the combination of colistin with several other agents compared with any agent alone [139,140]. The potential synergistic activity of polymyxin B with other antibiotics has been evaluated in seven studies, most of them against *A. baumannii* isolates [70]. Considering temocillin, moderate synergism was found in combination with other beta-lactam antibiotics. Moreover, the combination of temocillin with fosfomycin demonstrated beneficial in vitro and in vivo results against *E. coli* strains that produced carbapenemases [141]. The emergence of fosfomycin resistance was prevented with the addition of temocillin, and the combination proved to be more bactericidal that fosfomycin alone [142]. Fosfomycin with even meropenem or imipenem can act synergistically against *E. coli* and *A.baumannii* strains in preventing the emergence of fosfomycin resistance [143,144]. The degree of synergism required for a clinically effective combination is unknown, although one could intuitively assume that a clinically effective combination would result in MIC reduction below the corresponding ECOFF or susceptibility breakpoint of the first drug at clinically achievable concentrations of the second drug and vice versa. 

In conclusion, in silico modelling indicated clinical potentials of polymyxin B, temocillin, mecillinam, and fosfomycin against certain species of gram-negative MDR pathogens. Further studies are required in order to test the clinical efficacy of those antibiotics against MDR infections.

## Figures and Tables

**Figure 1 pharmaceuticals-15-01501-f001:**
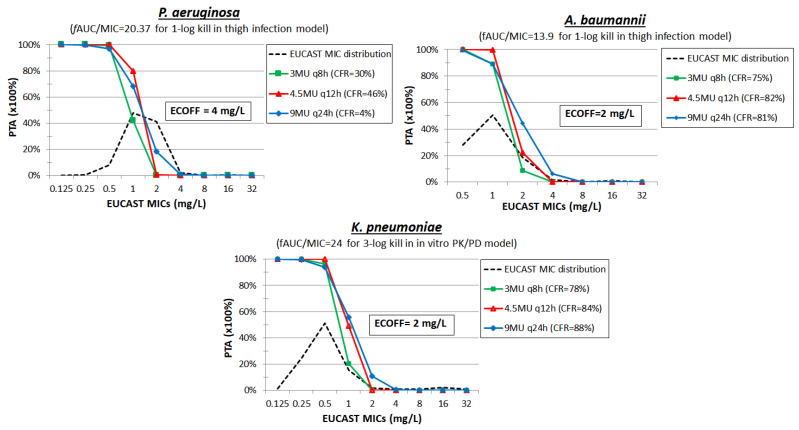
Probability of target attainment (PTA) of colistin at different dosing regimens against isolates with increasing MIC. The cumulative fraction of response (CFR) is shown for a collection of isolates with the EUCAST MIC distribution.

**Figure 2 pharmaceuticals-15-01501-f002:**
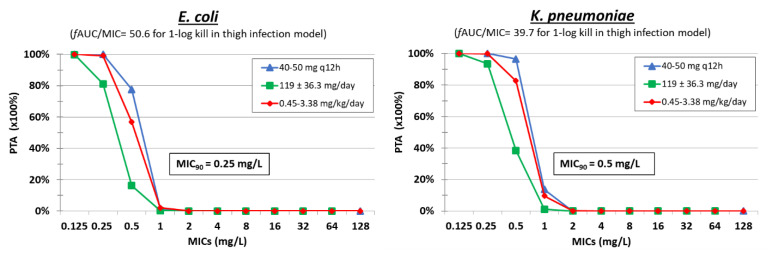
Probability of target attainment (PTA) of polymyxin B at different dosing regimens against isolates with increasing MIC.

**Figure 3 pharmaceuticals-15-01501-f003:**
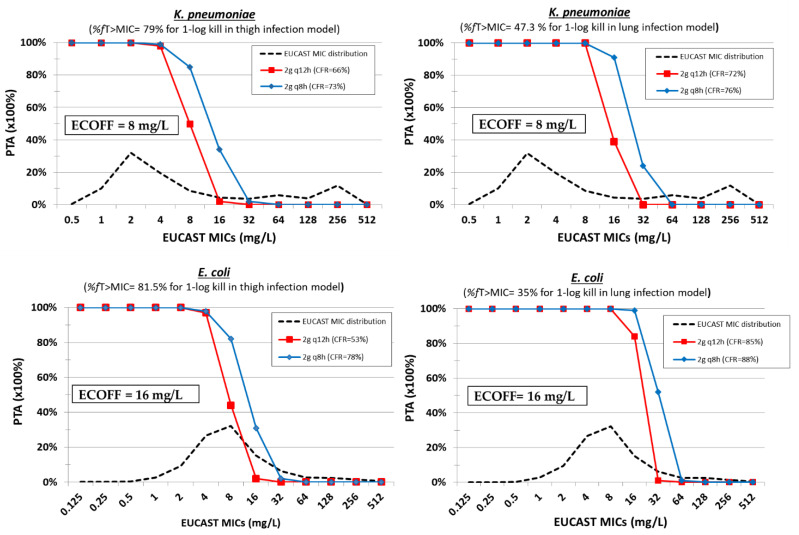
Probability of target attainment (PTA) of temocillin at different dosing regimens against isolates with increasing MIC. The cumulative fraction of response (CFR) is shown for a collection of isolates with the EUCAST MIC distribution.

**Figure 4 pharmaceuticals-15-01501-f004:**
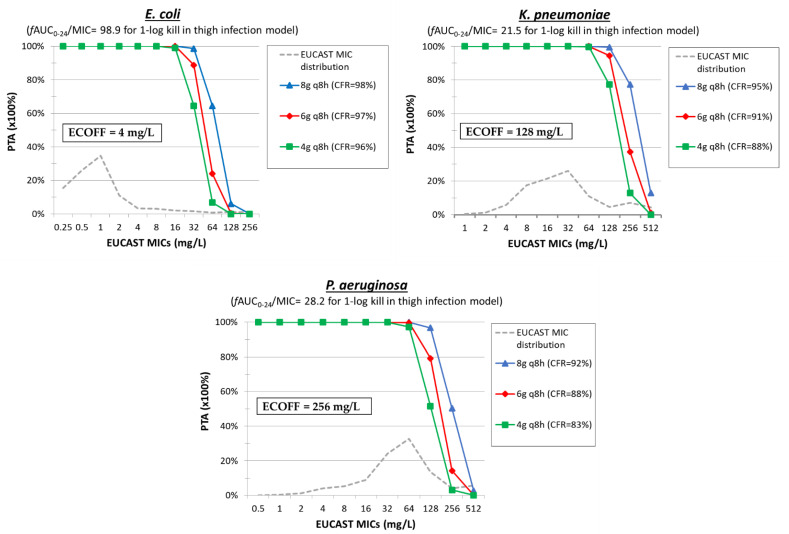
Probability of target attainment (PTA) of fosfomycin at different dosing regimens against isolates with increasing MIC. The cumulative fraction of response (CFR) is shown for a collection of isolates with the EUCAST MIC distribution.

**Figure 5 pharmaceuticals-15-01501-f005:**
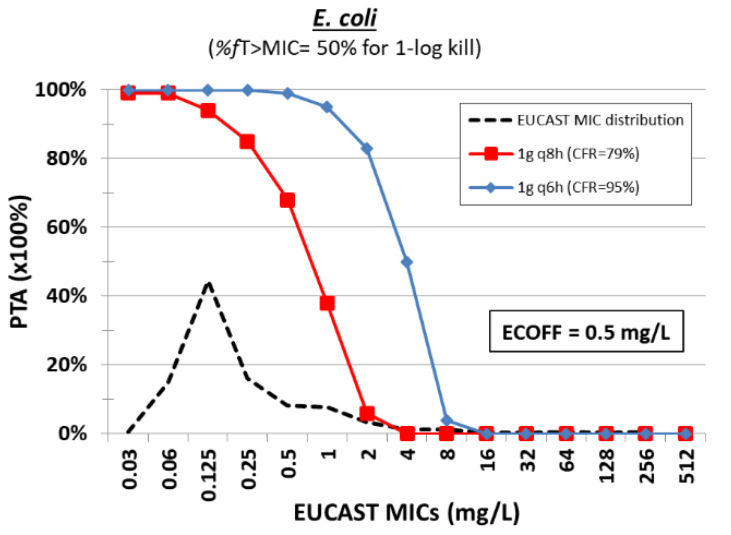
Probability of target attainment (PTA) of mecillinam at different dosing regimens against isolates with increasing MIC. The cumulative fraction of response (CFR) is shown for a collection of isolates with the EUCAST MIC distribution.

**Figure 6 pharmaceuticals-15-01501-f006:**
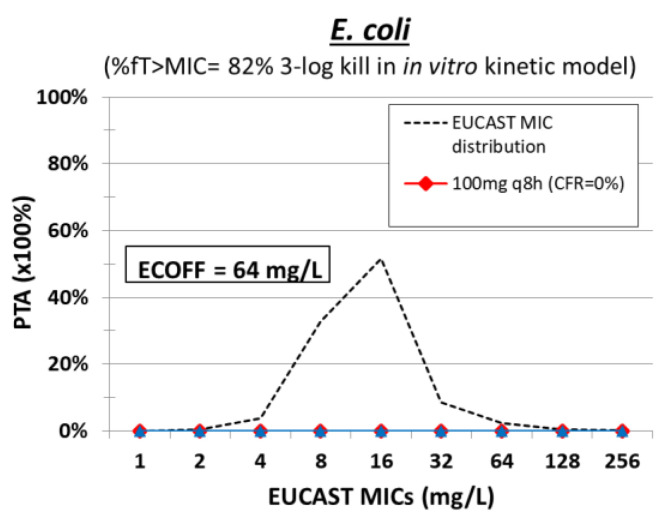
Probability of target attainment (PTA) of nitrofurantoin at different dosing regimens against isolates with increasing MIC. The cumulative fraction of response (CFR) is shown for a collection of isolates with the EUCAST MIC distribution.

**Figure 7 pharmaceuticals-15-01501-f007:**
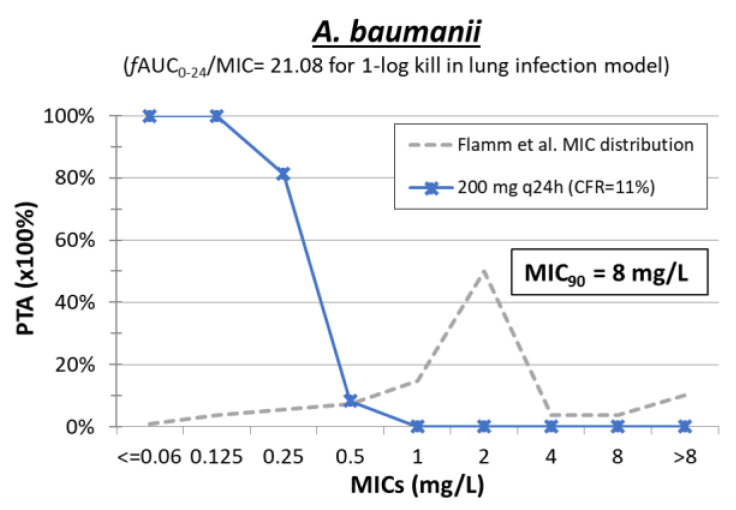
Probability of target attainment (PTA) of minocycline against isolates with increasing MIC. The cumulative fraction of response (CFR) for *A. baumannii* is shown for a collection of MDR isolates with MIC distribution by Flamm et al [37].

**Figure 8 pharmaceuticals-15-01501-f008:**
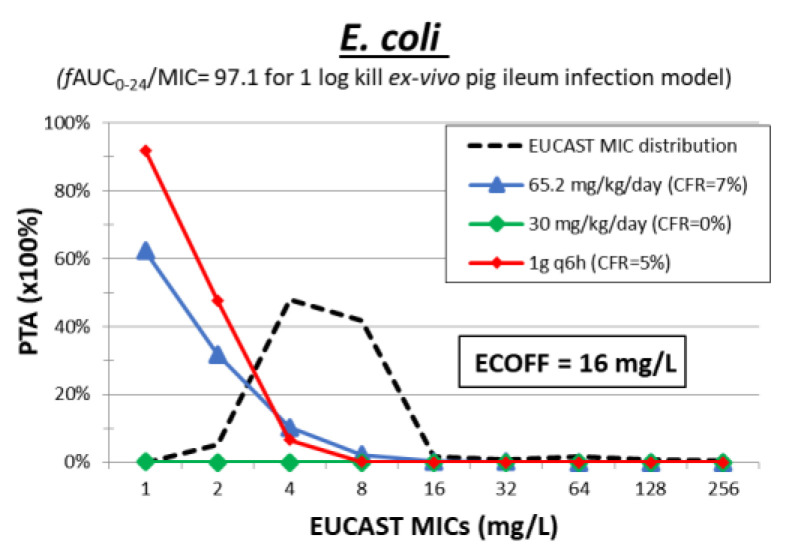
Probability of target attainment (PTA) of chloramphenicol at different dosing regimens against isolates with increasing MIC. The cumulative fraction of response (CFR) is shown for a collection of isolates with the EUCAST MIC distribution.

**Table 1 pharmaceuticals-15-01501-t001:** In vitro activity of old antibiotics against most common gram-negative bacteria.

Antimicrobial Compound	No. of Isolates (Ref.)	Mechanism ^b^	MIC Range (mg/L)	MIC_50_ (mg/L)	MIC_90_ (mg/L)	ECOFF ^c^
**Colistin**						
*E. coli*	6014 ^a^	NA	2–128	0.5	1	2
	715 [13]	ESBL	NA	≤0.25	0.5	
*K. pneumoniae*	1841 ^a^	NA	0.125–512	0.5	1	2
	21 [14]	CRKP	0.25–2			
	633 [13]	ESBL	NA	0.5	2	
*P. aeruginosa*	19,270 ^a^	NA	0.06–128	1	2	4
	698 [13]	MDR	NA	1	2	
*A. baumannii*	2879 ^a^	NA	0.5–32	1	2	[2]
	106 [15]	NA	0.125–32	0.5	1	
	75 [16]	CRAB	0.5–2	1	2	
**Polymyxin B**						
*E. coli*	17,035 [17]	NA	NA	≤0.5	≤0.5	
	86 [18]	All including ESBL	0.064–8	0.125	0.25	
*K. pneumoniae*	96 [19]	CRE	0.5–16	2	2	
	173 [20]	CRE	≤0.125–>64	2	32	
	186 [18]	All including ESBL	0.064–4	0.25	0.5	
*P. aeruginosa*	8705 [21]	NA	≤1–>8	≤1	2	
	124 [22]	CRPsA included	NA	1	2	
*A. baumannii*	18 [23]	OXA-23	0.06–0.5	0.25	0.25	
	131 [18]	All including CRAB	0.125–>32	0.25	0.5	
**Temocillin**						
*E. coli*	5702 ^a^	NA	0.125–512	8	32	16
	105 [24]	ESBL	2–32	8	16	
	162 [25]	ESBL	1–64	8	16	
	198 [26]	CTX-M	0.125–>128	8	16	
	293 [27]	CTX-M	2–64	8	16	
	40 [27]	AmpC	2-64	8	16	
*K. pneumoniae*	605 ^a^	NA	0.5–512	2	256	8
	23 [28]	Non-CTX-M ESBL	4–32	16	32	
	199 [27]	CTX-M	2–64	8	32	
*P. aeruginosa*	104 [29]	NA	64–≥256	≥256	≥256	
*A. baumannii*	51 [29]	NA	2–≥256	≥256	≥256	
**Fosfomycin**						
*E. coli*	2351 ^a^	NA	0.125–512	1	4	4
	35 [30]	ESBL	≤0.5–1024	1	32	
	24 [31]	KPC, MBL, OXA-48	≤0.25–256	1	256	
	528 [32]	ESBL	NA	NA	2	
*K. pneumoniae*	1396 ^a^	NA	0.5–512	32	256	128
	50 [30]	KPC	8–>2048	32	>2048	
	27 [30]	NDM, OXA-48	16–>2048	512	>2048	
	50 [31]	KPC, MBL, OXA-48	0.5–>1024	16	256	
*P. aeruginosa*	701 ^a^	NA	1–512	64	128	256
**Mecillinam**						
*E. coli*	1502 ^a^	NA	0.03–512	0.125	2	[0.5]
	198 [26]	CTX-M	0.125–>128	1	8	
	30 [33]	Resistance to C3G		0.5	4	
	29 [34]	NDM	0.5–32	4	8	
*K. pneumoniae*	175 ^a^	NA	0.03–512	0.25	128	
	24 [34]	NDM	2–>32	8	>32	
**Nitrofurantoin**						
*E. coli*	4000 ^a^	NA	1–256	16	16	64
	105 [24]	ESBL	2–512	16	64	
	528 [32]	ESBL	NA	NA	64	
**Minocycline**						
*E. coli*	1498 ^a^	NA	0.125–64	1	16	4
*K. pneumoniae*	938 ^a^	NA	0.125–64	2	16	8
	70 [35]	KPC	0.06–64	4	16	
	164 [36]	CP-Kp	NA	8	32	
*A. baumannii*	539 [37]	MDR	0.06–8	2	>8	
	401 [37]	XDR	0.06–8	2	>8	
**Chloramphenicol**						
*E. coli*	45,852 ^a^	NA	1–256	4	8	16
*K. pneumoniae*	65 ^a^	NA	1–256	4	8	
*A. baumannii*	202 [38]	MDR	≤2–≥32	≥32	≥32	

^a^ Data from www.eucast.org, accessed on 23 October 2022. ^b^ Abbreviations: NA = Not Available information, mostly WT, MDR = Multiple Drug Resistance, ESBL = Extended Spectrum Beta-Lactamase, CRKP = Carbapenem-resistant *K. pneumoniae*, CRAB = Carbapenem-resistant *Acinetobacter baumannii*, CRE = carbapenem-resistant *Enterobacterales*, CRPsA = Carbapenem-resistant *Pseudomonas aeruginosa*, OXA = Oxacillinase, KPC = *Klebsiella pneumoniae* carbapenemase, MBL = *Metallo*-*β*-*lactamase*, NDM = New Delhi metallo beta lactamase, AmpC = Ampicillinase C. ^c^ Tentative ECOFF are in brackets. Cells are left empty when no data are available.

**Table 2 pharmaceuticals-15-01501-t002:** Clinical pharmacokinetics of old antibiotics.

Drug	Dosage Regimen	Patient Population (Ref.)	CL (L/h)	VD in L (Mean ± SD)	% Unbound	AUC_0–24_ (mg·h/L) (Mean ± SD)	Cmax (mg/L) (Mean ± SD)	t1/2 (h) (Mean ± SD)
**Colistin**	3 MU q8h	13 ICU patients [43]			40%	50.18 ± 10.74	3.34 ± 0.35	7.8 ± 0.76
	4.5 MU q12h	13 ICU patients [43]			40%	60.71 ± 12.0	2.98 ± 0.27	8.8 ± 0.55
	9 MU q24h	13 ICU patients [43]			40%	72.93 ± 38.57	5.83 ± 0.87	9.6 ± 0.62
**Polymyxin B**	40–50 mg q12h	50 renal transplant patients [44]	1.18 ± 0.1	12.09 ± 1.58		74.6 ± 17.81	8.15	
	119 ± 36.3 mg q12h–q24h	35 adult patients [45]	2.5 ± 1.1	34.3 ± 16.4		52.3 ± 14.8/ 45.1 ± 17.3		10.1
	0.45–3.38 mg/kg q12h–q24h	24 critically ill patients [46]	1.4	V1 = 6.3, V2 = 23.1	42	66.9 ± 21.6	2.79 ± 0.90 ^d^	11.9
**Temocillin**	2g q12h	10 ICU patients [47]	2.44 ± 0.39	14.3 ± 0.87	23.7 ± 6.15	1856 ± 282	147 ± 12	4.3 ± 0.3
	2g q8h	14 critically ill patients [48]	3.69 ± 0.45	V1 = 14 ± 2.51 V2 = 21.7 ± 4.52	41	1764	170	
	0.5g	10 healthy volunteers [49]	1.5 ± 0.09	10.5 ± 0.7	12	344.1 ± 18.7	77.9 ± 28.4	5.2 ± 0.3
	1 g	10 healthy volunteers [49]	1.78 ± 0.08	11.9 ± 0.7	14	573.3 ± 27.8	160.8 ± 58.2	5.0 ± 0.2
	2g	10 healthy volunteers [49]	2.62 ± 0.16	16.8 ± 0.7	37	784.5 ± 47.1	236.1 ± 93.3	5.0 ± 0.2
**Fosfomycin**	4g q6h	16 non-critically ill [50]	2.43 ± 1.64	13.69 ± 2.81	100 ^c^	5215.08 ± 1972.2	422.6 ± 86.8	
**Mecillinam**	400 mg	9 subjects [51]			90–95 ^c^	22 ± 5	28 ± 5	
	200 mg	9 subjects [51]			90–95 ^c^	9.9 ± 1.5	12 ± 2	
	10 mg/kg	12 healthy volunteers [52]	14.7 ± 1.4	16.1 ± 2.8	90–95 ^c^		61	0.85 ± 0.14
**Nitrofurantoin** **(oral)**	50 mg q6h	12 healthy adult female [53]	36.4 ± 11.4	100.0 ± 49.6	25–50 ^c^	4.43 ± 0.96	0.326 ± 0.081	2.3 ± 1.8
	100 mg q8h	12 healthy adult female [53]	46.2 ± 18.6	103.8 ± 65.9	25–50 ^c^	6.49 ± 2.9	0.69 ± 0.35	1.7 ± 0.6
**Minocycline**	200 mg	55 critically ill patients [54]	5.24 ± 2.63	146 ± 57	30 ± 12	24.3 ± 7.88	2.58 ± 1.33	*T*_1/2,α_ = 1.36 ± 0.456 *T*_1/2,β_ = 23.4 ± 9.53
**Chloramphenicol** **Sodium succinate**	65.2 (32.3–114.4) mg/kg/day	10 critically ill patients [55]	21.24 ± 23.34	21 ± 8.4	~40 *^a^	468 ± 498		1.20 ± 1.15
	30 mg/kg	7 patients [56]	22.08 ± 10.32	133 ± 56	34–63 *^b^	72 ± 32	16.2 ± 9.1	
	1 g q6h	8 patients [57]	7.72 ± 1.87	23.1 ± 9.1	~40 *^a^	518	8.4-26.0	0.57 ± 0.12

*^a^ Based on Burke et al., J Pharmacokinet Biopharm. 1982, 10, 601–614. *^b^ Based on DOI: 10.2165/00003088-198409030-00004. ^c^ Based on EUCAST rationale document. ^d^ Css,avg. Cells are left empty when no data are available.

**Table 3 pharmaceuticals-15-01501-t003:** Plasma PK/PD target (mean ± SD) for old antibiotics.

Drug	Infection Model (Ref.)	Dose	Species	PK/PD INDEX	Stasis	1-Log Kill	2-Log Kill	3-Log Kill
**Colistin**	Thigh [58]	5–160 mg/kg/day q3h–q24h	*P. aeruginosa* (n = 3) MIC = 0.5–1 mg/L	*f*AUC/MIC	13.35 ± 4.57	20.37 ± 4.13	31.63 ± 4.27	58.37 ± 7.27
	Lung [58]	5–160 mg/kg/day q3h–q24h	*P. aeruginosa* (n = 3) MIC = 0.5–1 mg/L	*f*AUC/MIC	5.31 ± 1.18	14.83 ± 2.35	40.13 ± 5.01	127 ± 19.29
	In vitro [39]	*f*Cmax 9, 3 and 1.5 mg/L q8h–q24 h	*K. pneumoniae* MDR(n = 2) MIC = 0.5–2 mg/L	*f*AUC/MIC	10 ± 2.1	14 ± 2.4	18 ± 3.1	24 ± 4.7
**Polymyxin B**	Thigh [59]	4–512 mg/kg q6h, q12h, q24h	*E. coli* MDR (n = 4) MIC = 1 mg/L	*f*AUC/MIC	63.5 ± 34.8	50.6 ± 3.8		
	Thigh [59]	4–128 mg/kg q6h, 8–256 mg/kg q12h, 256– 512 mg/kg q24h	*K. pneumoniae* MDR (n = 5) MIC = 0.5–2 mg/L	*f*AUC/MIC	11.6 ± 22.1	39.7 ± 14.4		
	Thigh [60]	0.5–120 mg/kg/day	*K. pneumoniae* (n = 3) MIC = 0.25–1 mg/L	*f*AUC/MIC	6.73 ± 6.23	16.37 ± 12.17		
**Temocillin**	Thigh [61]	8–512 mg/kg q2h, 16–512 mg/kg q4h	*E. coli* ESBL (n = 4) MIC = 8–16 mg/L	%*f*T > MIC	66 ± 9.9	81.5 ± 14.4		
			*K. pneumoniae* ESBL (n = 4) MIC = 8–64 mg/L	%*f*T > MIC	63 ± 27.9	79 ± 6.4		
	Lung [61]	16–1024 mg/kg q2h, 32–1024 mg/kg q4h	*E. coli* ESBL (n = 4) MIC = 8–16 mg/L	%*f*T > MIC	27.8 ± 13.8	35 ± 18.3	42.8 ± 23	
			*K. pneumoniae* ESBL (n = 4) MIC = 8–64 mg/L	%*f*T > MIC	35.8 ± 23.6	47.3 ± 21.4		
**Fosfomycin**	Thigh [62]	12.5–6400 mg/kg/day q3h–q24h	*E. coli* ESBL (n = 5) MIC = 1–16 mg/L	*f*AUC_0–24_/MIC	23.7 ± 15.3	98.9 ± 78.4		
	Thigh [62]	12.5–6400 mg/kg/day q3h–q24h	*K. pneumoniae* NDM (n = 3) MIC = 4–16 mg/L	*f*AUC_0–24_/MIC	11.1 ± 19.5	21.5 (n = 1)		
	Thigh [62]	12.5–6400 mg/kg/day q3h–q24h	*P. aeruginosa* (n = 2) MIC = 8–16 mg/L	*f*AUC_0–24_/MIC	14.6 ± 4.7	28.2 ± 17.82		
**Mecillinam**	EUCAST RD		*Enterobacterales*	%*f*T > MIC	30–35%	~50%		
**Nitrofurantoin**	In vitro kinetic model [63]	16 mg/L	*E. coli* (n = 1) MIC = 2 mg/L	%*f*T > MIC	72%			82%
**Minocycline**	Lung [64]	0.46–180 mg/kg/day q12h	*A. baumannii* (n = 6) MIC = 0.03–4 mg/L	*f*AUC_0–24_/MIC	13.75 ± 3.76	21.08 ± 7.24		
**Florfenicol** *	Ex-vivo pig ileum [65]	30 mg/kg Single dose	*E. coli* (n = 1) MIC = 8 mg/L	AUC_0–24_/MIC	82.83 ± 3.52	97.1 ± 4.12		101.6 ± 7.74

* Drug class of Chloramphenicol PK/PD target used. The PK/PD target of florfenicol was determined in the ileum fluid. Cells are left empty when no data are available or corresponding targets are not reached.

**Table 4 pharmaceuticals-15-01501-t004:** Summary of old antibiotics with clinical potential against gram (-) isolates.

Old Antibiotics	*E. coli*	*K. pneumoniae*	*P. aeruginosa*	*A. baumannii*
Colistin	-	-	-	-
Polymyxin B	✓	✓	-	✓
Temocillin	✓	✓	-	-
Fosfomycin	✓	✓	-	-
Mecillinam	✓	-	-	-
Minocycline	-	-	-	-
Nitrofurantoin	-	-	-	-
Chloramphenicol	-	-	-	-

✓ attain preclinical PKPD targets, - do not attain preclinical PKPD targets.

## Data Availability

Data sharing not applicable.

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
