# Peer review of "Assessing Clinical Potential of Old Antibiotics against Severe Infections by Multi-Drug-Resistant Gram-Negative Bacteria Using In Silico Modelling"

_pharmaceuticals, 2022, doi:10.3390/ph15121501_

Round 1

Reviewer 1 Report

Dear authors.

The problem of bacterial resistance to many bacteria is relevant for the whole world. "Super-bacteria" (resistant to all known antibiotics) is no longer a new concept. But "the new is the well–forgotten old" does not work with the microcosm. The old antibiotics and the people who took them had their effect on the resistance of microorganisms. They have formed a genetic memory. I don't see much point in such studies. To conduct a modeling is great, but then empirically confirm the ineffectiveness of old antibiotics. Then what is the novelty of the acquired knowledge? The authors need to answer these questions.

There are also other comments. 

Major comments:

1) L.69, "the probability of target attachment", what is the target attachment?

2) Figures 1-8, authors need to change the axis signature and the caption. Probability is a number that takes values from 0 to 1.

3) Table 4, notation is not clear.

4) Conclusion. The main idea of this study has been formulated. The old antibiotics enhanced the effects of the new ones in some cases. It is necessary to study the synergistic effect of several antibiotics against severe infections.

5) In the References, 38% of publications refer to 2018-2022 (the last 5 years); the remaining 62% of used sources are older than 5 years. It is recommended to increase the share of references to sources published over the last 5 years when analyzing the current state of research in the area under consideration, since this area of knowledge is rapidly developing.

Author Response

We thank the reviewers for his critical assessment of our work. We have addressed all his comments, point by point.

Reviewer #1

The problem of bacterial resistance to many bacteria is relevant for the whole world. "Super-bacteria" (resistant to all known antibiotics) is no longer a new concept. But "the new is the well–forgotten old" does not work with the microcosm. The old antibiotics and the people who took them had their effect on the resistance of microorganisms. They have formed a genetic memory. I don't see much point in such studies. To conduct a modeling is great, but then empirically confirm the ineffectiveness of old antibiotics. Then what is the novelty of the acquired knowledge? The authors need to answer these questions.

Response: The novelty of the current review is that it summarize all in vitro MIC, PK, PK/PD and PTA data in order to assess the clinical potential of old antibiotics to treat MDR infections. The current review indicated that there is a potential for some drugs against certain species and it provides hypotheses that need to be tested clinically. This is now stated in page 15.

Major comments:

1) L.69, "the probability of target attachment", what is the target attachment?

Response: Changed to target attainment.

2) Figures 1-8, authors need to change the axis signature and the caption. Probability is a number that takes values from 0 to 1.

Response: Title of X axes has been changed to probability of target attainment (x100%).

3) Table 4, notation is not clear.

Response: Notations have been revised.

4) Conclusion. The main idea of this study has been formulated. The old antibiotics enhanced the effects of the new ones in some cases. It is necessary to study the synergistic effect of several antibiotics against severe infections.

Response: As the focus of the present review was to assess the clinical efficacy of these antibiotics against infections caused by MDR pathogens with in silico modeling, main purpose of the present study didn’t expand on synergistic interactions  of old antibiotics with others (newer or older). Moreover, information about antibiotic synergism effect can be found in DOI: 10.1080/14787210.2020.1705155, where authors review the potency of monotherapies versus combination therapy in some of old antibiotics included in the present study.

5) In the References, 38% of publications refer to 2018-2022 (the last 5 years); the remaining 62% of used sources are older than 5 years. It is recommended to increase the share of references to sources published over the last 5 years when analyzing the current state of research in the area under consideration, since this area of knowledge is rapidly developing.

Response: Most studies on old antibiotics have been conducted many years back. Hence a large proportion of references are older than 5 years. We have used some more recent data but those papers are limited.

Reviewer 2 Report

The manuscript entitled: In Silico Modelling Assessing Clinical Potential of Old Antibiotics Against Severe Infections by Multi-Drug Resistant Gram-negative Bacteria is investigating an interesting topic however, consider the following commnets:

1- The title should be simplified and clarified

2-More details should be added to the abstract

3- The abbreviated words should be mentioned complete for the first time in the manuscript, then their abbreviation should be mentioned.

4- Please clarify why these antibiotics were selected among the other old antibiotics.

5- MIC90 should be clarified

Author Response

We thank the reviewer for his critical assessment of our work. We have addressed all their concerns, point by point.

Reviewer #2

1- The title should be simplified and clarified

Response: Title was rephrased.

2-More details should be added to the abstract

Response: More details have been added to the abstract.

3- The abbreviated words should be mentioned complete for the first time in the manuscript, then their abbreviation should be mentioned.

Response: Abbreviations have been revised.

4- Please clarify why these antibiotics were selected among the other old antibiotics.

Response: The selection of mentioned antibiotics, was based on previous papers indicating the potential for using those antibiotics to treat MDR infections (DOI: 10.1093/jac/dkv157). As there was a great interest on those antibiotics, we focus our review on those.

5- MIC90 should be clarified

Response: MIC90 has been clarified in page 3.

Round 2

Reviewer 1 Report

Dear Authors 

My comments are taken into account